# From Analyses to Implementation and Innovation

**Daniel P. Loucks**

Cornell University, Ithaca, New York, NY 14853–0001, USA; loucks@cornell.edu or dpl3@cornell.edu

**Abstract:** Reviews of the water resource systems planning and management literature show considerable interest in methodological issues and less so in implementation experiences. This paper offers some thoughts on the use of our analysis tools in the political environment where water management decisions are typically made. This paper also addresses the challenge of going beyond analysis and synthesis to innovation. How can we extend our modeling methods so as to help ourselves become more creative in the identification of potentially improved infrastructure design and/or operating policies, and even of institutional changes, that we have otherwise not considered or thought of?

**Keywords:** systems analyses; water resources; planning; management; implementation; political processes; innovation; impact

---

## 1. Introduction

This commentary is addressed to all of us who develop and apply various quantitative modeling approaches designed to assist those responsible for managing water and related environmental resources. This includes those of us trained in various disciplines that offer different perspectives and contribute to identifying and analyzing alternative solutions in different ways, all aimed at forming a more comprehensive estimate of the impacts that could result from decisions that might be made.

Those of us who have been involved in the use of systems analysis methods are aware of the contribution these methods have made and are making in a wide range of applications, including agriculture, defense, ecosystem management, education, environmental protection, industry, law enforcement, medical care, resources management, transportation, and urban planning among others. Systems analyses have been most helpful in addressing issues dominated by natural and physical sciences and engineering. Yet such issues are usually addressed and resolved in a political environment. This is certainly the case when planning, designing, and operating infrastructure for managing water. This paper focuses on the implementation of systems analysis for informing the largely political processes of deciding how best to manage our water resources. If the purpose of our analyzing specific water resource systems is to implement change, then we, analysts, must get involved in and cater to the political processes in which water management decisions are typically made.

Most of us will agree that while systems analysis methods, and each of the disciplines they come from, have their limitations, they can introduce a certain objectivity into the political process of decision making. Progress in managing water more effectively requires knowledge from the natural, social and political sciences, economics and other disciplines. Of course, achieving change requires institutions and political alignments in addition to the insights derived from scientific knowledge. Yet such scientific objectivity can help achieve stakeholder acceptance of the identified options available and the inevitable tradeoffs among the goals they may wish to obtain [1]. Such analyses can address uncertainties, even uncertain uncertainties; they can estimate various impacts and tradeoffs among multiple system performance measures; and they can help reveal unexpected consequences of particular policies and actions. We can use systems analysis methods to help identify plans and policies that achieve a balance

among multiple goals of multiple stakeholders. Simply stated, systems analysis methods have proven themselves to be useful for addressing large, complex water management challenges and opportunities. Results from such analyses can inform but, with rare exceptions, they have not proven very effective in substituting for those responsible for decision making.

The art of applying systems analysis tools, especially to water management issues, is itself inherently multi- and interdisciplinary. One can argue that it was borne in a multidisciplinary environment [2]. Systems analysis approaches are designed to focus on the performance of entire systems rather than each of their components, but just what components are or are not included in a particular water resource system depends on the management issues being addressed and the authorities given to the institutions involved. If the art of defining the system components and their interactions is done well, and in collaboration with those involved in decision making so as to enhance communication, gain trust, and ensure relevance, there is an excellent chance that the structured and objective nature of the systems approach will provide information considered useful by those involved in the decision-making process [3–5].

## 2. Mismatches

Water resource management issues arise when there are mismatches between what people want or desire and what they are getting or observing. There seems to be a continuing stream of such mismatches reported in the news media each day. They give proof that many of our water management problems have become very large and very complex, technically and politically, and that these mismatches can have substantial adverse consequences on our wellbeing as individuals and as communities, and also on our environment. Addressing and reducing these mismatches is a challenge given the uncertainties in supplies and demands. Without the aid of our analysis tools, it is considerably more difficult to deal with such problems simply by, but not excluding, intuition or hunches [6].

Consider some headlines that have made the news in recent months as reported in Circle of Blue <info@circleofblue.org>:

- Utilities in Colorado, US, prepare for water shortages amid the lowest mountain snowfall in 30 years.
- Volatile weather patterns cause rivers across Germany to overflow their banks.
- Southeast England may be at risk of water shortages following a year of dry weather.
- More than 200 flood alerts were in place across the UK, including several severe or "danger to life" warnings. Fifteen rivers across England's Midlands, Yorkshire, and Lancashire have reached their highest levels ever recorded, and an estimated 3300 English homes have been flooded. Several hundred homes in Wales were inundated as well.
- Somalia experiences its fourth consecutive failed rainy season, exacerbating the country's instability.
- Disputes between Texas and Colorado and New Mexico over the Rio Grande and between Florida and Georgia over the allocation of the water flowing from the Blue Ridge Mountains are being addressed by the U.S. Supreme Court.
- Water shortages play a role in ongoing unrest across Iran.
- Drought, flooding, and other natural disasters threaten half of U.S. military bases worldwide.
- Taps have been on the verge of running dry in several major global cities, including Cape Town, South Africa; Mexico City, Mexico; Melbourne, Australia; and Kabul, Afghanistan. the United Nations claims this will happen to 2/3rds of the globe by the year 2025.
- Almost one-fifth of the world's population, live in areas of physical scarcity, and 500 million people are approaching this situation.
- Almost one quarter of the world's population face economic water shortage due to inadequate infrastructure.

- Last year, the Mekong river's waters dropped to the lowest in a century. The water has changed to an ominous color and begun filling with globs of algae. Fish in the Mekong, the world's largest inland fishery, are emaciated.
- Glacier melt in western China increases, threatening the water supply of 1.8 billion people
- Tests results following a massive fish die-off in Iraq's Euphrates River show high levels of bacteria and heavy metals in the waterway.
- U.S. food trade increasingly depends on groundwater use that is not sustainable.
- Flooding and landslides in Belo Horizonte, Brazil, have killed over 50 people.
- Chemicals, including pesticide DDT, are found in the tissues of dolphins swimming in waters flowing to the Great Barrier Reef.
- Heavy flooding in Madagascar displaces at least 16,000 people.
- A vessel runs aground on the Danube river in northern Bulgaria due to low water levels, blocking a key shipping route.
- Ongoing research reveals the pervasiveness of polyfluoroalkyl substances (PFAS). These "forever chemicals" are estimated to be in the bloodstream of 99 percent of Americans, and some scientists believe that nearly all of the country's surface water is likely contaminated.

The list could go on. What is clear is that there are many places and times where widespread mismatches between the desired flows, levels, and qualities of water and what exists. The question is what to do about issues such as these. It is the responsibility of water managers to address these issues, and one way of identifying, analyzing, and evaluating alternative options is through the use of systems analysis methods. Yet such analyses by themselves will not change anything. To effect change, one has to perform such analyses in collaboration with those institutions having the responsibility and authority to make water management changes in specific situations. Analysts need to address the goals (as stated and as understood) of these institutions, recognizing that these goals can change during the time analyses are being performed. Lawyers are useful participants in such efforts. They can translate the results of our systems analyses into the legislation needed to enable changes. Skillful analysts are those who can work in a multidisciplinary environment that may include engineers, economists, ecologists, lawyers, planners, and politicians among others.

## 3. Water Resources Systems Analysis

No doubt everyone reading the papers in this series knows what systems analysis is, but I have to admit that when I began studying this subject, no one knew much about what that term meant, except for the fact that our military had a bunch of so-called "whiz kids" using systems analysis methods to 'win' the Vietnam war. (Clearly, systems analysis has its limitations!) I began studying this subject just as the Harvard Water Program published their first book [2] showing how optimization and simulation models running on computers could be used to address water resources management issues in ways that integrated economics, hydrology, engineering and political science perspectives. Pretty neat and pretty exciting!

Since then, we have been busy developing and applying many different types of modeling methods, each having its strengths and weaknesses. So far, we have not found one best modeling approach, and I am convinced that we will not. What we have been able to do because of improvements in both model solution algorithms and computer technology is to address increasingly more complex and comprehensive water resources management issues using a variety of methods. From the perspective of a scientist and researcher, a primary role of systems analysis approaches is to contribute to a better understanding of real-world water system performances, humans included, and how they can be improved. From the perspective of a water manager, the primary role of systems analysis methods is to provide quantitative information to help them do their job, i.e., support their decision-making processes [7].

Much of our water resources systems literature today focuses on new modeling approaches (the hammers), often selecting data from particular rivers or basins or urban areas (the nails) to illustrate how their hammers perform. This literature rarely addresses actual model implementations in an institutional environment. However, that does not mean those implementations are not taking place. Firms such as Danish Hydraulics Institute (DHI) and Deltares, and government agencies such as US Environmental Protection Agency (USEPA) and US Army Corps of Engineers-Hydrologic Engineering Center (USACE-HEC) here in the United States are heavily engaged in the implementation of their models and software. The same applies for developers and users of Aquatool [8], CalSim [9], IRAS [10], Riverware [11], WEAP [12,13], and other models used for planning and even real-time operation. Experiences using these models are rarely written up and published in professional journals so most of us cannot learn from those experiences. While I know all of us in this business of modeling water resource systems enjoy inventing new hammers and applying them, going the extra step of actually using them in a political decision-making environment is, in my view, even more fun, more challenging, and certainly very educational. We model developers would all benefit if more of these experiences were included in our literature, including the periodic reviews or assessments of the state of the art of water resources systems modeling such as in [14–37].

## 4. Implementing Systems Analysis

So, what about the use of systems analysis in support of institutions involved in addressing water management mismatches and making changes in the way water is managed? If I had to summarize my experiences using systems analysis within decision-making processes, I would have to admit that while the results of modeling almost always helped focus the debates on what decisions to make, the decisions themselves were not exactly as I would have predicted. The relative importance of various objectives or even the objectives themselves almost always changed during the planning and decision-making processes, and sometimes even immediately after those process ended. (For example, one month after the completion of a national water resources infrastructure development plan, the country's president died, and the new president and his new department ministers decided to discard that three-year effort carried out by a previous administration, giving the study to others do over again.) My conclusion based on my limited experiences over some five decades is that one should expect such surprises and be ready to adapt to them. All this is in part why, at least for me, these experiences have never been boring and indeed have taught me more than I could have imagined when I began studying for this profession [38].

The motivation to use systems analysis is to identify how to make something better, i.e., how to reduce mismatches. To the extent that the results of the analyses are implemented and improvement actually happens as predicted by the analyses is one measure of success. However, it is not the only one in my opinion. A more achievable measure of success is the extent that the results of systems analyses influences the debate on what to do to achieve improvement. An analysis can be superb technically but if no one pays any attention to its results when debates are taking place about what to do, I will judge it as being unsuccessful with respect to implementing change. Admittedly, it may make a great journal paper. If it has an impact on our research in systems analysis methods, it can certainly be considered successful with respect to that goal.

To affect change in how water is managed, however, we need to do our modeling and analyses so as to have buy in. For this, we need to produce results that are not only deemed useful, and timely, but also simultaneously enhance the salience, credibility, and legitimacy of the insights that they produce. To accomplish this requires, at a minimum, staying in close contact with those involved in decision making throughout the decision-making process. Further, that is not always easy. As just mentioned, goals, constraints, system boundaries, and even stakeholders can change during the period of analysis. An awareness of institutional goals and constraints is critical, and again these will likely change. Stated objectives may differ from what is really desired. Stakeholders and decision makers can change over time, and thus their goals may change. They may really not know what they will

want until they know better what they can have and do, perhaps informed by information coming from various analyses taking place over time. Two-way communication between us, analysts, and our clients needs to be maintained throughout the period of developing, using and solving models if we hope to be useful in influencing the debate about what decisions to make.

## 5. Infrastructure

Water management is accomplished through the design and operation of water infrastructure that permits us to alter the temporal and spatial distribution of water and its quality and the benefits derived from the various uses of it. Infrastructure can include water and wastewater distribution and collection systems, treatment plants, surface and subsurface storage, pumps, canals, aqueducts, cisterns, rain gardens, flood protection measures, and facilities for generating hydropower, cooling, navigation, and rainwater harvesting. Most analyses of water resource systems are focused on addressing what, if any, infrastructure to develop and/or operate, where (siting), when (staging), to what extent (capacity), and why.

Water infrastructure can provide important benefits to society, but it can also generate adverse impacts as well. Today, the flows of water and sediment in over two-thirds of the world's major rivers are altered by dams, diversions, and levees. Close to 1000 new dams are planned or under construction just in South America, in Southeast Asia, and in China. This expansion of dams and associated infrastructure is driven by the need to better satisfy agricultural, domestic and industrial demands for reliable water supplies, for more energy, for recreational opportunities, for reduced risks of damages from droughts and floods, among other purposes. However, dams and levees, for example, can alter the geomorphology of rivers and the functioning of wetland ecosystems including fish habitats, and downstream deltas. Due to population increases and accompanying increases in demands for water, many more dams are being built and for sure, along with their benefits will be their adverse impacts [39].

Our water resources systems analysis literature is full of papers exploring the use of particular methods for identifying, analyzing and evaluating infrastructure and policies for, for example, responding to floods, droughts, and other catastrophic events, restoring ecological habitats, preventing pollution, meeting domestic and industrial water supply and water quality demands, generating power, providing recreational opportunities, meeting energy and agricultural demands, and informing and educating the public on water issues. Many of these water management issues are driven by a changing climate that is bringing us more frequent and more intense storms, floods, droughts, and corresponding land erosion and pollution [40]. Adding to these stresses on our water resource systems are the increasing demands for adequate, reliable, clean and inexpensive supplies of water and for reducing the discharges of a wide range of pollutants that are threating human and ecosystem health. Our early literature mostly focused on models with economic objectives and constraints. However, even if estimated net benefits may have to be positive, it seems to me that many agencies managing water today are less interested in improving economic efficiency or effectiveness as in minimizing the chances of being criticized for doing the wrong thing and in making sure they are spending the money allocated and available to them. This suggests we need to acknowledge not only economic and hydrologic uncertainty but also institutional and political behavior. We often tend to beat to death the former, because we can, and ignore the latter, because it is much harder. It is hard to know what political/institutional uncertainties to consider when we do not even know what they could be, let along their probabilities. It is hard to guess how objectives and constraints, and indeed human behavior, may change over time, and what future generations, many of whom are not yet alive, will want us to do for them as we develop infrastructure today that will exist for a long time. Further, when we do think we know what future generations would want from us, such as reduced greenhouse gas emissions, some alive today will object. There are no 'optimal' solutions no matter what goals are used to rank alternatives. We often have to settle for any change in policies that seems feasible and compatible with how implementing institutions work and that stakeholders will support. I know all of this is not new

to anyone involved in practicing the art of water resources systems analysis or engineering. However, it reinforces the argument on how essential it is to perform these analyses in collaboration with our clients if we expect to inform and even influence their decisions. Decisions related to water resources planning and management are made by people in their institutional environments, not by models or algorithms, especially those developed and solved without client involvement.

## 6. Challenges

It is always fashionable to use the word 'complex' when describing the systems we are analyzing. Nevertheless, is fair to say that many of the water resource systems being analyzed in recent years really are complex. It is one thing to use optimization for identifying the dimensions of a least-cost n-sided (n ≥ 3) water storage tank as part of an urban water storage and distribution network. This is not a complex problem. However, interestingly, the solution always shows that the minimum total cost results when two-thirds of the tank's total cost is associated with its sides. Knowing this means that we do not need to use optimization models for identifying the dimensions of least-cost tanks. Having such rules of thumb is rare for the systems we are typically asked to study. Without the use of systems analysis methods, it is hard to imagine how else we would estimate the values of all the design and operating decision variables and resulting performance measures we need to be able to improve system design and operation.

Consider, for example, part of the water distribution system for the US city of Houston, in Texas. Houston is a small city compared to some of the world's largest cities, but it is still complex. Its water and sewer system covers a service area of some 640 square miles and consists of over 7500 miles of drinking water and over 6500 miles of wastewater networks. Issues facing the managers of this system include handling the risk of being overwhelmed by floods and having to flush untreated wastewater into surface waters, thereby putting the public at risk of disease and infection. Prolonged power outages could further stress the system's ability to operate. Power outages could shut down some of the city's 380 pump stations. If there is no power, there is no water. In addition, over time, distribution networks break and when they do there is a risk of drinking water supplies becoming contaminated and people becoming sick. Every city in the world faces such challenges. Learning how to address such problems efficiently and effectively over time is clearly a complex challenge [41].

For another example, consider the US state of California. Hundreds of dams, many aquifers and pumping stations, and tens of thousands of kilometers of aqueducts, service a variety of agricultural, commercial, industrial, residential, environmental and ecological water demands throughout the state. These demands include water for instream flows, wetlands, and for maintaining cool or warm water temperatures depending on the local aquatic species of concern. This complex system is managed by numerous federal and state agencies and local water districts and suppliers. Some of these institutions have project management authority and others have regulatory authority. Their water management decisions impact millions of water users each day. They also impact the state's food, energy, industrial and public health sectors. One can be justified by calling even a small part of California's water resource system complex. Many within California have been involved in the development and implementation of models for analyzing and evaluating various alternative designs and operating policies of major parts of California's water resource infrastructure. They have done this in close collaboration with appropriate federal and state and local agencies and to their credit have had a beneficial impact on the management of the state's water [42,43].

These are just two examples of what I would call complex systems. They are made up of multiple interacting components, both physical and institutional, and have hundreds of decision variables that managers can assign values to when attempting to satisfy the multitude of objectives people want from the system. They must make their decisions under all the uncertainties of future water supplies and demands [44,45].

Adding to the complexity of many water resource systems are their links to the energy sector. The energy required to transport, clean, and heat water is estimated to consume approximately

13 percent of US's total national energy supply. As more communities turn to desalination and water recycling technologies the energy embedded in each unit of treated water will rise. At the same time, continued use of coal and nuclear material will further strain water supplies. The energy sector consumes more water for thermoelectric cooling than any other sector, at least in the United States. Rising energy use and prices will drive up the use and price of water and vice versa. Surely this will occur as the future demands for both water and energy increase.

Because of multiple challenges such as these, there has been a push toward using systems analysis methods to address how to develop and manage increasingly integrated water-energy-food-environmental resource systems. However, efforts to achieve greater integration often run up against the lack of institutions having overall responsibilities for managing water supply, wastewater treatment, stormwater management, flood control, energy and agriculture linkages, and habitat restoration especially if these issues cross political boundaries. The lack of such institutions does not negate the value of learning how better to manage water and related resources, including working with nature and thereby benefiting from sustainable ecosystem services that a healthy environment can provide. Who knows, perhaps such studies will motivate institutional cooperation if not change.

## 7. From Analysis to Innovation?

Systems analysis tools give us some useful tools for helping us plan, design, manage and operate more effective systems. However, they cannot generate suggestions on what we should also be thinking about in addition to what we have been thinking about and including in our models. Our analysis methods do not have the creative capacities that our human brains have. When computer technology became available for developing and using menu-driven graphics interfaces allowing direct interaction between the models and the model users, some of us got excited about the potential of directly involving stakeholders and decision makers in the analyses of various water resource systems. Many of us believed that generating pictures that could show the impact of various design and management decisions or assumptions they might want to explore, and providing the interactivity that would allow them to explore and get estimates of the impacts of different decisions they might make, would give them a better understanding of the system being modeled and how it might work. Such interactive visual displays would also let them explore how they might improve system performance, and indeed help them think about options that could be 'outside the box.' In other words, help them innovate.

We even got fancy with respect to performing sensitivity analyses and displaying uncertainty. Our displays were clear, understandable, and colorful. Sometimes, we witnessed users even believing what they were seeing, ignoring the fact that the models creating the graphical and pictorial displays were approximations of reality. It was fun developing and using such tools in a participatory environment [46–48]. Of course, today almost every model used to analyze water resource systems incorporate interactive, graphics-based, interfaces. They remain approximations of reality. (Sound and smell and touch are coming!)

But what we, modelers and analysts, have not done yet is to figure out how to make our models suggest planning and management options other than what they are programmed to consider. Right now, our models can only inform us about a system we have defined in some general way. To return to our simple tank example, any model for finding the dimensions of a tank is not going to suggest alternatives that might negate the need for that tank. Similarly, if we are modeling a proposed reservoir, say on the Mekong or Nile Rivers, in addition to learning how fast it may fill with sediment under different hydropower production and sediment management policies, would it not be nice if our model could suggest other sites, other designs and other options, such as the use of solar panels for generating power, that might be preferable to what has been modeled, and show the appropriate impacts of those panels regarding costs, power production and sediment and fish passage [49].

In short, models can analyze and synthesize but they cannot yet innovate. They cannot suggest different systems boundaries or components. They cannot suggest different components or policies or options that we have not already included in them. Yet humans can think and create new ideas

that are outside the scope of any particular model. How can we develop models that help humans do this? Some are exploring the use of games. Some are suggesting interactive evolutionary computation, maybe coupled to artificial intelligence (and even to the use of LSD). Also waiting to be used in more creative and innovative ways are massive data, Google search engines and their ability to access all the information available on the Internet, Google Earth, voice recognition (that kids take for granted when asking their cell phones questions), parallel cloud computing, and even three-dimensional virtual reality environments that stakeholders can step into and interact with [50,51]. (We can dream!).

I am not at all optimistic about our ability to model and predict human behavior, but with such enhanced interfaces, stakeholders—including decision makers—involved in the use of our computer-based models can predict how they might react to and behave given some possible future scenario. We, model builders and analysts, can then observe what questions are asked and what decisions are made by real humans before and during simulations of a water system. If nothing else, this information should improve our water resources planning and management in the future.

## 8. The Future

A crisis in water is no longer an abstraction for many. Climate change, underfunded and aging infrastructure, outdated or limited management approaches and institutional authorities, and the pressures of urbanization and the growing demands for food and energy are stressing water infrastructure worldwide. Aging infrastructure, growing populations and shifting patterns of settlements, and increasing costs are all making water management one of leading infrastructure challenges in many regions of our planet. A changing climate is skewing precipitation patterns that guided earlier engineering and making water scarcity and water-related natural disasters topics of concern almost everywhere.

Populations, especially in urban centers, are often impacted by inadequate water quality. The most serious clean water issues occur in water bodies where older combined sewers, diffuse non-point urban stormwater, and growing runoff of agricultural pollutants have remained largely uncontrolled. Improvements to many of our receiving waters in North America, such as Chesapeake Bay, the Everglades, the Great Lakes, the Gulf of Mexico, Puget Sound, and the San Francisco Bay-Delta, will require multistakeholder strategies using insights derived from the application of our best and most appropriate systems analysis tools.

Losses in water-related biodiversity continues, from pressures to divert critical water supplies to agriculture and urban uses, and from the loss of wetlands and other critical water habitats. Learning to use green infrastructure—and the services that such functioning ecosystems offer, from fisheries and recreation to water purification and flood protection—is increasingly gaining acceptance in water resource management agencies. Disputes over regulating base flow water levels essential to commercial and non-commercial fish species are common throughout much of the world.

Changes in our climate are now altering hydrologic cycles and impacting how water is being managed. The variability of water supplies has increased and hence the reliability of those supplies has decreased. These trends will no doubt continue into the future. Warmer temperatures and changing precipitation patterns are reducing annual snowpacks and increasing evaporation, reducing the performance of reservoirs and the services provided by watersheds. Increasing weather extremes will stress not only humans but also wildlife and natural systems. Sea level rise and higher intensity storms will steadily increase risks of coastal and inland flooding. Adapting to these impacts will challenge all of us. We face these challenges at a time when the limited financial resources available to pay for the needed water infrastructure calls for management decisions that are more cost and socially effective, perhaps in addition to being just politically feasible. We, analysts, should be able to contribute to this effort. Our most compelling contribution is our systematically gathered, analyzed and objectively interpreted information. Our job, as I see it, is to identify alternative plans, designs or policies, that are politically feasible and identify the tradeoffs, if any, with respect to what is expected or desired by all stakeholders. This includes effectively communicating this information to those engaged in the political decision-making process of policy implementation. The political environment in which

we use our analyses to inform water managers can dictate the choice of the systems analysis methods we use and how our technical support influences the decision-making process and its outcomes.

Finally, one can analyze the scientific aspects of water resource systems in increasing detail all day, every day, but if these quantitative analyses do not take into account the qualitative stakeholder, decision maker and institutional biases, emotions and opinions, the result will likely fail to influence how water will be managed. We, modelers, need to figure out how the information we produce can have a greater beneficial impact in the real, as opposed to just the published, world. The answer is not in paying attention to either the current research on modeling methods or on aspects that enhance real-world implementation and impact, it is in considering both of these activities together [52].

**Funding:** This research received no external funding.

**Conflicts of Interest:** The author declares no conflict of interest.

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
