# Peer review of "From Analyses to Implementation and Innovation"

_water, doi:10.3390/w12040974_

Round 1

Reviewer 1 Report

Thanks for giving me a precious opportunity to read through an interesting commentary written by an experienced civil engineering researcher. The argument presented in the commentary should appeal many practitioners and academic researchers who have been involved in modeling water resources management and exposed to a series of questions raised by the gap between modelings and the reality. 

Since I myself is a trans-disciplinary person based on social sciences, the message of the commentary sounds quite reasonable rather normal, but as I stated above, people in natural scientists must feel in a different way.

There are some comments/questions in a positive meaning:

  • Is it possible to introduce your understanding of "system" and "system analysis method" even though this commentary would be one of papers that would be issued as a special volume or something? It should be very helpful for readers to coherently understand the context.
  • It might be better to clarify whom this commentary is targeting: water resources managers, persons in political processes, natural scientists who are lost in the implicative gap or others.
  • The examples of mismatches are considerably difficult for me to understand why they are called as "mismatches."
  • The references mentioned in page 4 (line 151) are more than 20 only for one part. Some more explanations might be required for understanding the author's intention, or just decrease the numbers.
  • The linkage of the system analysis method and enhancing "an awareness of institutional goals and constraints" (p.4, line184) is not clear due to the lack of explanation about system analysis method itself. 
  • The conclusion (p.7, line 335-) makes sense.

Reviewer 2 Report

This paper highlights challenges that are faced around the world and I agree with most of the arguments put forward by the Author, and I would suggest these also apply more broadly as well to include down scale of GCM’s for predictions, etc. etc. etc.   The paper is the Authors personal views and for this review I’ve made some observations that may be worth reflection.

The first is to also ask the author if not only innovation but ‘impact’ is a key word.   In the ‘International Development’ global water resources sphere that I’ve considerable experience, we see a wide range of ‘innovation’ that ultimately cannot be scaled to a level that provide true ‘impact’.   We also as engineers and scientists may have insightful innovations which if implemented would have great impact, but fail due to lacking support from physical, economic, social or political infrastructure.  We therefore need to ask how to ultimately use ‘output’ (be it models etc.) and innovation to implement impact that meets long-term objectives.   

The second being the Sustainable Development Goals which are not named in the paper.    The items indicated for Circle of Blue  [email protected], are those which are the tip of an iceberg (and really don't need to be listed) and are there as someone has raised the profile of these problems (as pointed out by the Author). There are a vast number of issues that don’t get this attention in countries around the world, which impact society and peoples (local to national).  One of the drivers for the SDG’s is to raise the profile ‘in country’ and in ‘governing region’s’ to the challenges around governance, policy, regulation and management of resources (including water).  We need to engage the academic and practitioner community to develop policy-relevant and impactful studies, results and solutions that can help guide targets and associated policy changes, one of the goals of this paper no doubt.

Might it be helpful to acknowledge the global goals and what the authors thoughts are on their ambitions in respect of your arguments?

The third is around ‘modelling’, I have always taught my students (at all levels) that a model is a conceptual representation of the real world and that models only answer the questions you pose within your personal understanding of the system.   It is never 100% true or correct and the conceptualization must be reviewed when interpreting the results.     I agree with the Author that models, no matter how complex, cannot provide exact solutions for policy and management, this requires balancing of science/engineering within stakeholder engagement that ultimate informs the decisions made.   The model results should thus indicate the limitations and boundaries of their consideration, and importantly the academic and practitioner community needs to be clear on the policy ramifications of their work, and not only the innovation of ‘doing’ but the impact of the results – ie stick their necks out……  (like the conclusions from a recent paper in Water

Water 201911(10), 2160; https://doi.org/10.3390/w11102160

From Conclusions:   “The results for the Diyala Basin (Iraq) demonstrate that groundwater storage depletion was 25% over 10 years of water exploitation, increasing to about 60% over 25 years. The aquifer storage was completely exhausted after 40 years in both alternatives due to low aquifer recharge caused by low rainfall and high evapotranspiration rates (semi-arid zone). The introduction of drip irrigation mitigates the impact on the aquifer storage over the discrete periods, especially for long-term water exploitation. The probability of sustainable groundwater resource management was scenario modelled for the next half-century by reducing water delivery demands. The results show that a possible sustainable storage budget using an open furrows system can be achieved for the next 25 years, and 33 years for a drip system with 45% demand’s yield for both. Hence, the decision makers should consider future policy to reduce water demands by either changing crops types or reducing farms areas. Also, the use of a drip system for water allocation should be considered in policy since it has less impacts on groundwater yields. Conjunctive use with surface water may also be considered, to mitigate groundwater depletion and maintain its sustainability.”

The final point is around the political dimension.  Lines 161 – 164 provides an anecdote by the Author, perhaps following with a more concrete discussion of the reality that science-led policy instruments are also open to political scrutiny, and it is all too often that clear objectives from an engineering/science perspective are not palatable at the political levels.  The author could point the readers to say (for example) the OECD review of water governance that continues to share best practice, and perhaps this might be one conduit that supports nations to advise leadership to use modelling as the Author indicates?

https://www.oecd.org/env/watergovernanceprogramme.htm

In conclusion, the paper adds to a global debate and should provide the readers with an experienced  perspective for cogitation.

Specific Notes:

Line 118-120 is not needed.
